# Wine Yeast Cells Acquire Resistance to Severe Ethanol Stress and Suppress Insoluble Protein Accumulation during Alcoholic Fermentation

Masashi Yoshida,[a] Noboru Furutani,[a] Futsuki Imai,[a] Takeo Miki,[b] Shingo Izawa[a]

[a]Laboratory of Microbial Technology, Graduate School of Science and Technology, Kyoto Institute of Technology, Kyoto, Japan
[b]Laboratory of Food Manufacturing and Microbiology, Graduate School of Life and Environmental Sciences, University of Yamanashi, Kofu, Yamanashi, Japan

**ABSTRACT** Under laboratory conditions, acute 10% (vol/vol) ethanol stress causes protein denaturation and accumulation of insoluble proteins in yeast cells. However, yeast cells can acquire resistance to severe ethanol stress by pretreatment with mild ethanol stress (6% vol/vol) and mitigate insoluble protein accumulation under subsequent exposure to 10% (vol/vol) ethanol. On the other hand, protein quality control (PQC) of yeast cells during winemaking remains poorly understood. Ethanol concentrations in the grape must increase gradually, rather than acutely, to more than 10% (vol/vol) during the winemaking process. Gradual increases in ethanol evoke two possibilities for yeast PQC under high ethanol concentrations in the must: suppression of insoluble protein accumulation through the acquisition of resistance or the accumulation of denatured insoluble proteins. We examined these two possibilities by conducting alcoholic fermentation tests at 15℃ that mimic white winemaking using synthetic grape must (SGM). The results obtained revealed the negligible accumulation of insoluble proteins in wine yeast cells throughout the fermentation process. Furthermore, wine yeast cells in fermenting SGM did not accumulate insoluble proteins when transferred to synthetic defined (SD) medium containing 10% (vol/vol) ethanol. Conversely, yeast cells cultured in SD medium accumulated insoluble proteins when transferred to fermented SGM containing 9.8% (vol/vol) ethanol. Thus, wine yeast cells acquire resistance to the cellular impact of severe ethanol stress during fermentation and mitigate the accumulation of insoluble proteins. This study provides novel insights into the PQC and robustness of wine yeast during winemaking.

**IMPORTANCE** Winemaking is a dynamic and complex process in which ethanol concentrations gradually increase to reach >10% (vol/vol) through alcoholic fermentation. However, there is little information on protein damage in wine yeast during winemaking. We investigated the insoluble protein levels of wine yeast under laboratory conditions in SD medium and during fermentation in SGM. Under laboratory conditions, wine yeast cells, as well as laboratory strain cells, accumulated insoluble proteins under acute 10% (vol/vol) ethanol stress, and this accumulation was suppressed by pretreatment with 6% (vol/vol) ethanol. During the fermentation process, insoluble protein levels were maintained at low levels in wine yeast even when the SGM ethanol concentration exceeded 10% (vol/vol). These results indicate that the progression of wine yeast through fermentation in SGM results in stress tolerance, similar to the pretreatment of cells with mild ethanol stress. These findings further the understanding of yeast cell physiology during winemaking.

**KEYWORDS** *Saccharomyces cerevisiae*, winemaking, severe ethanol stress, protein quality control, acquired resistance, alcoholic fermentation, insoluble proteins, proteostasis

Address correspondence to Shingo Izawa, thioredoxin@kit.ac.jp.

The authors declare no conflict of interest.

The budding yeast *Saccharomyces cerevisiae* is commonly used to produce alcoholic beverages owing to its excellent ethanol production capacity. Ethanol produced by yeast cells inhibits the growth of harmful microorganisms and creates a favorable environment for yeast cells with high ethanol tolerance. However, high concentrations of ethanol are toxic even to yeast cells. Severe ethanol stress impairs the activity of transporters and endocytosis (1–3), induces oxidative stress and changes in cellular membranes (4–7), and strongly represses nuclear export of the bulk poly(A)$^+$ mRNA and protein synthesis in yeast cells under laboratory conditions (8, 9).

Similar to severe heat shock, acute ethanol stress (10% vol/vol) also damages proteins, causing the accumulation of denatured insoluble proteins in yeast cells (10). Denatured protein accumulation leads to cytotoxicity and has been implicated in the pathogenesis of neurodegenerative diseases such as Alzheimer's disease and Parkinson's disease (11, 12). Protein quality control (PQC) systems in proteostasis networks play critical roles in determining the toxicity of denatured proteins. PQC systems primarily degrade or regenerate denatured proteins in eukaryotic cells using molecular chaperones, the ubiquitin-proteasome system, and autophagy (13–18). Furthermore, excessive amounts of denatured proteins are collected and sequestered at intracellular deposition sites, such as IPODs (insoluble protein deposits), INQs/JUNQs (intranuclear quality/juxta-nuclear quality control compartments), and cytosolic aggregates (CytoQs), to minimize toxicity (19–22). Therefore, yeast cells treated with acute 10% (vol/vol) ethanol stress and heat shock at 42℃ induce the formation of denatured protein deposition sites (23). In *S. cerevisiae*, Hsp42 and Btn2, together with Sis1, play important roles as aggregases (sequestrases) in the formation of deposition sites (10, 24–29). Aggregated proteins at deposition sites are efficiently disaggregated by the bi-chaperone system, mainly consisting of Hsp104, Hsp70 (Ssa1 to 4), and Fes1, for subsequent regeneration or degradation (13, 29–32).

Additionally, pretreatment with mild ethanol stress (6% vol/vol) activates the expression of PQC-related factors. We found that the expression of proteins comprising the bi-chaperone system (Hsp104, Hsp70, and Fes1), Hsp42, and Sis1 is upregulated in yeast cells during pretreatment with 6% (vol/vol) ethanol and is maintained under subsequent severe ethanol stress (23). It is conceivable that the increased expression of PQC-related factors in the pretreated cells partly contributes to the suppression of insoluble protein accumulation under subsequent severe ethanol stress.

In the typical fermentation process of wine and Japanese sake, ethanol concentrations in grape must and sake mash (*moromi*) gradually increase and eventually exceed 10% (vol/vol). Therefore, yeast cells in the fermentation process are gradually and continuously exposed to mild-to-severe ethanol stress as the ethanol concentration increases. This suggests two possibilities regarding insoluble protein levels in yeast cells during fermentation. Protein denaturation and accumulation of insoluble proteins may occur in yeast cells exposed to high ethanol concentrations. Alternatively, yeast cells may become resistant to the cellular impact of ethanol and suppress the accumulation of denatured insoluble proteins during the gradual increase in ethanol concentration, even at the late stage of fermentation.

To clarify these possibilities, we conducted alcoholic fermentation tests using wine yeast cells (EC1118 and OCM-2) and a synthetic grape must (SGM; Instituto Superior de Agronomia synthetic grape must) (33). The levels of denatured insoluble proteins were analyzed throughout the fermentation process. Insoluble protein accumulation was negligible in wine yeast cells, even when SGM ethanol concentrations exceeded 10% (vol/vol). Conversely, insoluble protein accumulation was induced when fermented SGM was collected and fed into wine yeast cells cultured under laboratory conditions. The findings suggest that wine yeast cells acquire stress resistance during the fermentation process and can suppress the accumulation of insoluble proteins. This study provides novel insights into yeast physiology during winemaking.

## RESULTS

**Ethanol stress response of wine yeast cells under laboratory conditions.** Prior to the fermentation test, we compared the ethanol susceptibilities of wine yeast strains

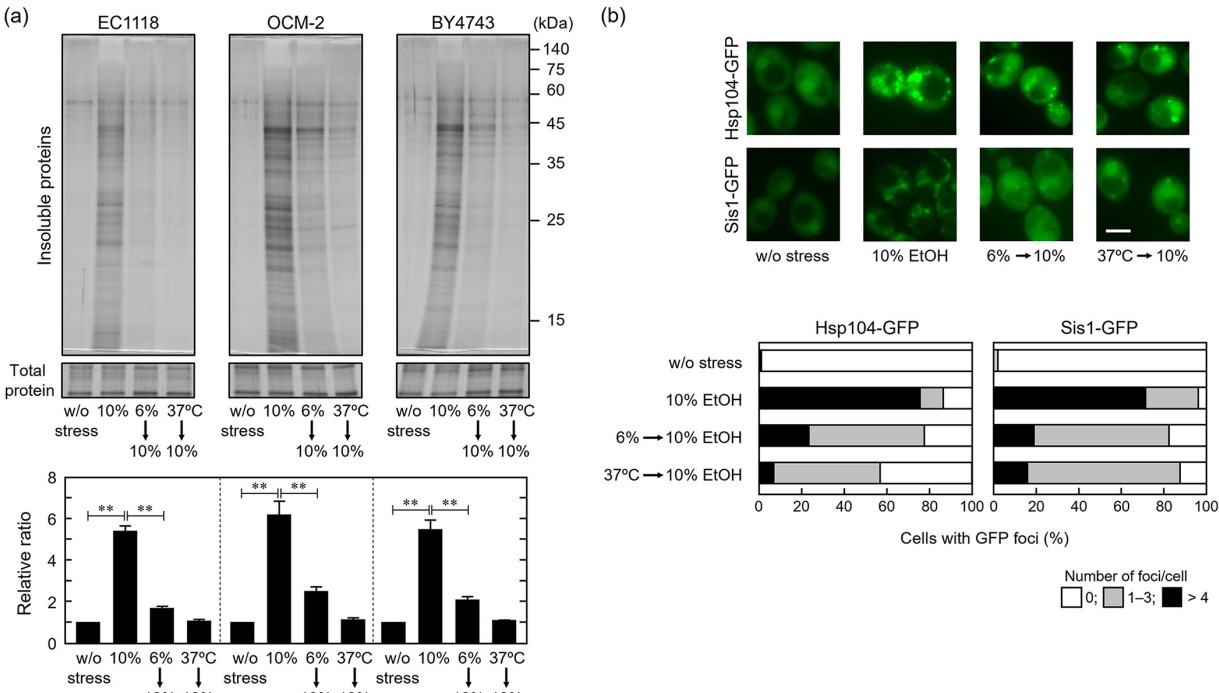

**FIG 1** Ethanol-induced protein denaturation of wine yeast cells under laboratory conditions. Cells were cultured in SD medium at 28°C with shaking (120 rpm). Log-phase cells ($OD_{600}$ = 0.5) were pretreated with or without mild stress (6% [vol/vol] ethanol for 180 min or at 37°C for 60 min) and then exposed to 10% [vol/vol] ethanol at 28°C for 180 min. All experiments were performed in SD medium with shaking (120 rpm). (a) Intracellular insoluble protein levels were assayed using silver staining. Each value is expressed as the mean $\pm$ standard error (SE) of fold changes in the staining levels of insoluble proteins, compared to cells without stress treatment ($n$ = 3). (b) Formation of deposition sites was examined using Hsp104-GFP and Sis1-GFP expression in OCM-2 cells. Representative images are shown in the upper panels and quantified data in the lower panels. A total of 100 cells were examined under each condition, and experiments were conducted in triplicate (300 cells in total were examined). Scale bar = 5 $\mu$m.

EC1118 and OCM-2, and the BY4743 diploid laboratory yeast strain under laboratory conditions. The levels of insoluble proteins under 10% (vol/vol) acute ethanol were determined in the strains. Yeast cells were cultivated aerobically in synthetic defined (SD) medium at 28°C. Insoluble protein accumulation was observed in EC1118, OCM-2, and BY4743 cells under 10% (vol/vol) ethanol stress. No significant differences were observed among the three strains under laboratory conditions (Fig. 1a). We also investigated whether the accumulation of insoluble proteins was suppressed in wine yeast cells under 10% (vol/vol) ethanol after pretreatment with mild stress, as previously demonstrated in haploid laboratory yeast cells (23). Exposure to 6% (vol/vol) ethanol for 180 min or 37°C for 60 min significantly reduced the insoluble protein levels under subsequent 10% (vol/vol) ethanol exposure in all three strains (Fig. 1a). The findings indicate that wine and laboratory yeast cells became resistant to severe ethanol stress with respect to insoluble protein accumulation under laboratory conditions.

Acquired resistance to severe ethanol stress through exposure to mild stress was confirmed by the mitigated formation of Hsp104-GFP and Sis1-GFP foci under 10% (vol/vol) ethanol. Hsp104-GFP is often used as a marker for the deposition sites of denatured proteins because Hsp104 binds to denatured protein aggregates and forms granules (24, 34–36). Sis1 is involved in the formation of deposition sites in the nucleus and cytoplasm and associates with them under conditions that cause protein denaturation (23, 25–29). To monitor the formation of the deposition sites, we expressed Hsp104-GFP and Sis1-GFP in OCM-2 cells and examined their intracellular localization. Direct exposure to 10% (vol/vol) ethanol induced the formation of multiple Hsp104-GFP and Sis1-GFP foci. Pretreatment with 6% (vol/vol) ethanol or thermal stress at 37°C significantly mitigated the formation of foci under subsequent 10% (vol/vol) ethanol stress in OCM-2 cells (Fig. 1b).

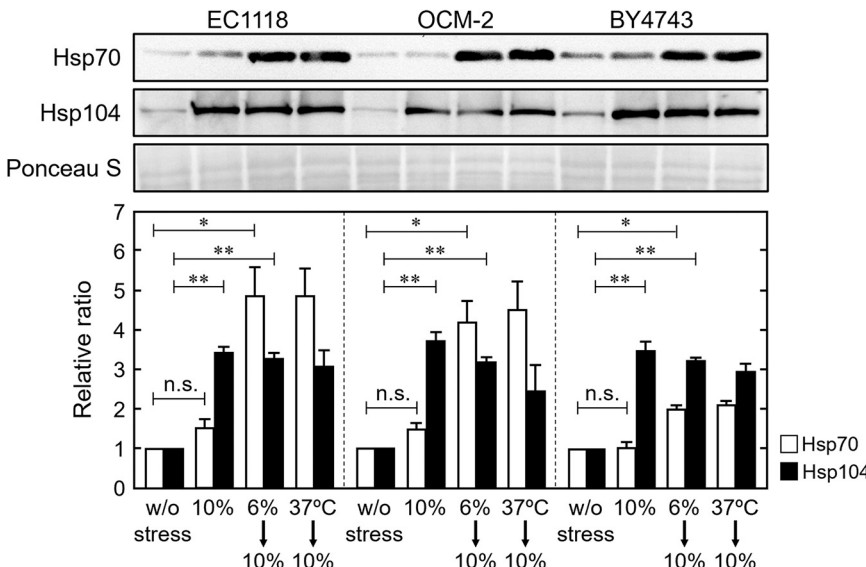

**FIG 2** Wine and laboratory yeast cells responded similarly to ethanol under laboratory conditions. Cells were prepared and exposed to stress, as described in the legend of Fig. 1. The expression levels of Hsp70 (white bars) and Hsp104 (black bars) were assayed using Western blotting. Each value is expressed as the mean ± SE of the fold changes in the protein levels relative to cells without stress treatment ($n = 3$).

We further examined the expression of Hsp70 and Hsp104 proteins in wine yeast cells. The proteins are key components of the bi-chaperone system (31, 32). Hsp104 expression was upregulated by severe ethanol stress in wine and laboratory yeast cells (Fig. 2). Additionally, pretreatment with 6% (vol/vol) ethanol or thermal treatment at 37°C increased Hsp70 levels under subsequent 10% (vol/vol) ethanol stress. No significant differences in the susceptibility or acquisition of ethanol resistance between wine and laboratory yeast cells were evident under laboratory conditions.

**Negligible insoluble protein accumulation in wine yeast cells during fermentation.** Fermentation tests were conducted at 15°C using wine yeast cells and SGM (33) to mimic white winemaking. EC1118 and OCM-2 cells showed increased ethanol concentration in SGM, as shown in Fig. 3a. The intracellular accumulation of insoluble proteins was negligible during most of the fermentation process in wine yeast cells (Fig. 3b). Additionally, the majority of cells formed a small number of Hsp104-GFP and Sis1-GFP foci in SGM, indicating that the formation of denatured protein deposits was less active during fermentation (Fig. 3c). Thus, insoluble proteins were maintained at low levels in wine yeast cells during the fermentation process, even when the SGM ethanol concentration exceeded 10% (vol/vol).

Further, we tested whether the levels of Hsp70 and Hsp104 could be increased during the fermentation process, as they were increased by the pretreatment with 6% (vol/vol) ethanol under laboratory conditions (Fig. 2). No significant induction of Hsp70 expression was observed during the process of fermentation (Fig. 4). Although Hsp70 levels increased slightly when EC1118 cells were transferred from SD medium to SGM, they decreased to around basal levels during fermentation. The expression of Hsp104 was increased following transfer of cells from SD medium to SGM. The increased expression was maintained during fermentation. However, Hsp104 levels in wine yeast cells in SGM were lower than those in SD medium with ethanol stress.

Hsp70, a major stress-responsive protein, did not increase significantly during alcoholic fermentation in SGM. We suspected that this was due to reduced stress from the fermented SGM. To verify this, yeast cells cultured under laboratory conditions were subjected to SGM during the fermentation process, and insoluble protein levels were examined. Fermented SGM (days 14 to 16) containing ~10% (vol/vol) ethanol did not induce

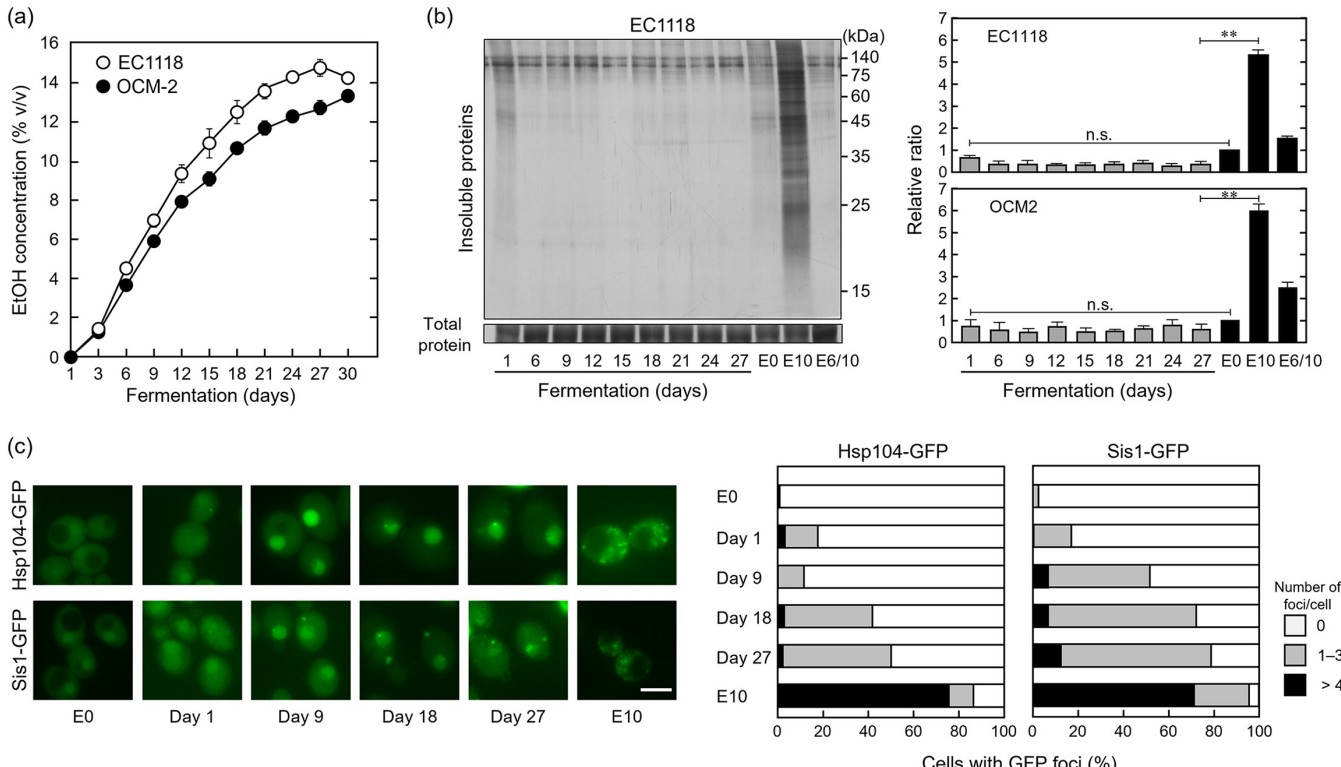

**FIG 3** The winemaking-like fermentation test was performed at 15°C without shaking using wine yeast strains (EC1118 and OCM-2) and synthetic grape must (SGM). (a) Changes in the SGM ethanol concentration during fermentation (*n* = 3). (b) Insoluble protein levels in wine yeast cells during fermentation. Cells were sampled on the days indicated during the fermentation process, and the cellular insoluble protein levels were examined. Day 1 samples were collected 6 h after yeast cells were transferred from SD medium to the SGM. As controls, log-phase cells cultured in SD medium were treated without stress (E0), exposed to 10% (vol/vol) ethanol for 180 min (E10), or pretreated with 6% ethanol for 180 min and then exposed to 10% (vol/vol) ethanol for 180 min (E6/10) in SD medium. Each value is expressed as the mean ± SE of fold changes in the staining levels of insoluble proteins relative to E0 (*n* = 3). (c) The formation of Hsp104-GFP and Sis1-GFP foci in OCM-2 was monitored during the fermentation process. Representative images are shown in the left panels and quantified data are presented in the right panels. A total of 100 cells under each condition were examined, and experiments were conducted in triplicate (300 cells in total were examined). Scale bar = 5 μm.

insoluble protein accumulation in wine yeast cells responsible for alcoholic fermentation (Fig. 3a and b). However, feeding fermented SGM (containing 9.8% and 11.8% ethanol) to wine yeast cells cultured in SD medium under laboratory conditions induced significant accumulation of insoluble proteins (Fig. 5). These results confirmed that SGM produced a stressful environment for yeast cells cultured under laboratory conditions after the middle stage of the fermentation process. Interestingly, fermented SGM with 9.8% (vol/vol) ethanol was more effective in causing protein denaturation than SD medium with 10% (vol/vol) ethanol. There may be unclarified factors within the fermented SGM that promote protein denaturation.

**Wine yeast cells become resistant to high-concentration ethanol during fermentation.** Considering that fermented SGM is a stressful environment, wine yeast cells exposed to alcoholic fermentation in SGM might become more resistant to stress. To assess this, we transferred wine yeast cells from fermenting SGM to fresh SD medium containing the same ethanol concentration to determine whether the accumulation of insoluble proteins was induced. Insoluble protein accumulation was suppressed when wine yeast cells from the fermentation process were treated with SD medium containing ~10% (vol/vol) ethanol (SGM to SD) compared with yeast cells cultured only in SD medium (SD) (Fig. 6). Thus, wine yeast cells in the fermenting SGM became more resistant to ethanol stress than wine yeast cells grown in SD medium.

Yeast cells do not actively grow in the later fermentation stage, with a state resembling the stationary phase (37). Additionally, cells in the stationary phase are more resistant to

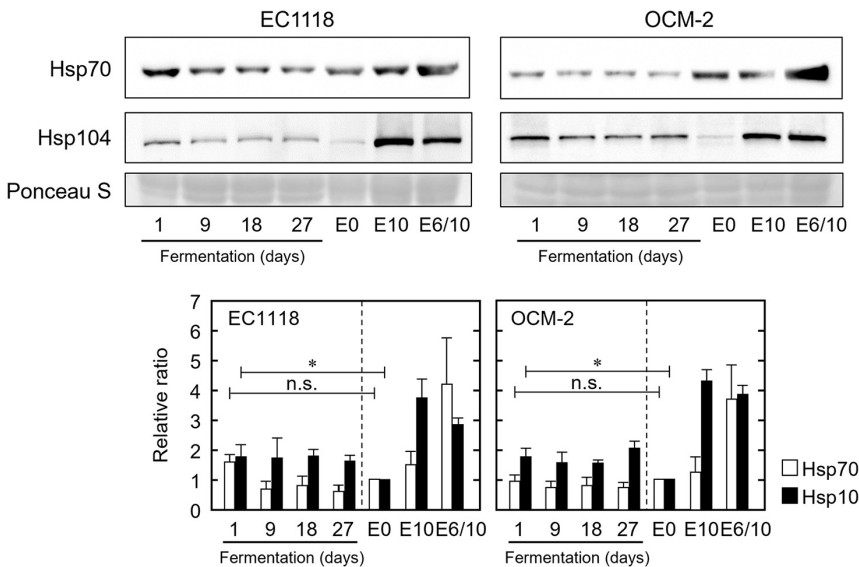

**FIG 4** Hsp70 and Hsp104 levels in wine yeast cells during fermentation were assayed using Western blotting. Samples were prepared as described in the legend of Fig. 3. As controls, log-phase cells cultured in SD medium were treated without stress (E0), exposed to 10% (vol/vol) ethanol for 180 min (E10), or pretreated with 6% ethanol for 180 min and then exposed to 10% (vol/vol) ethanol for 180 min (E6/10) in SD medium. Each value is expressed as the mean $\pm$ SE of fold changes in protein levels relative to E0.

various stressors than those in the logarithmic (log) growth phase (38, 39). We compared the accumulation of insoluble proteins caused by 10% (vol/vol) ethanol in stationary-phase (SP) cells and log-phase (LP) cells under laboratory conditions to determine whether SP cells suppressed the accumulation of insoluble proteins. Insoluble protein accumulation did not differ significantly between SP cells and LP cells exposed to 10% (vol/vol) ethanol (Fig. 7). The findings suggest that the improved tolerance of yeast cells during the fermentation process was not attributed to the stationary phase of growth.

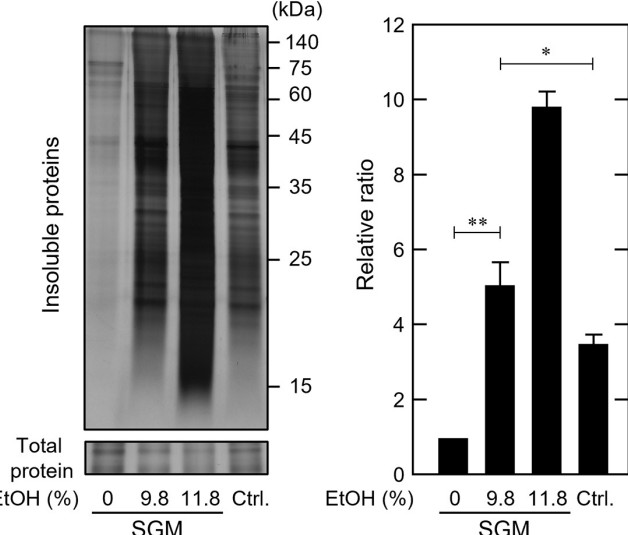

**FIG 5** Fermented SGM caused protein denaturation in yeast cells cultured under laboratory conditions. The SGM during fermentation performed by EC1118, which contained 9.8% (vol/vol) and 11.8% (vol/vol) ethanol, was collected and exposed to EC1118 cells cultured in SD medium for 180 min. EC1118 cells were also exposed to fresh SGM (0% ethanol). As a control (ctrl.), EC1118 cells cultured in SD medium were treated with SD medium containing 10% (vol/vol) ethanol for 180 min. Intracellular insoluble protein levels were assayed using silver staining. Each value is expressed as the mean $\pm$ SE of fold changes in the staining levels of insoluble proteins relative to cells exposed to fresh SGM ($n = 3$).

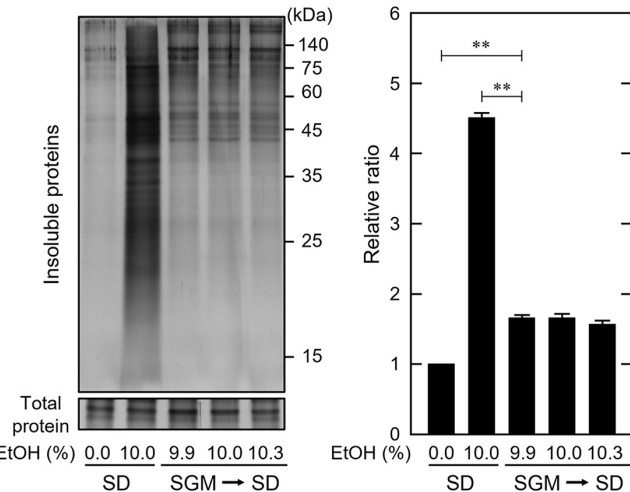

**FIG 6** Wine yeast cells became tolerant of the cellular impact of ethanol during fermentation. EC1118 cells in the fermentation process were collected from the SGM and treated with fresh SD medium with the same ethanol concentration for 180 min (SGM to SD). Samples were prepared from three independent fermentation processes, followed by determination of ethanol concentrations. As a control, cells cultured in SD medium were treated with SD medium with or without 10% (vol/vol) ethanol for 180 min (SD). The intracellular insoluble protein levels of EC1118 cells were then assayed. Each value is expressed as the mean ± SE of fold changes in the staining levels of insoluble proteins relative to cells exposed to SD medium without ethanol (*n* = 3).

## DISCUSSION

Insoluble protein accumulation was rarely observed in wine yeast cells during fermentation, even when SGM ethanol concentrations exceeded 10% (vol/vol). Additionally, active formation of denatured protein deposition sites was not observed throughout the fermentation process. Conversely, fermented SGM induced insoluble protein accumulation in wine yeast cells grown in SD medium, confirming late-stage fermenting SGM as a stress-inducing environment. Our results indicate that wine yeast cells become resistant to the cellular impact of severe ethanol stress and can suppress the accumulation of insoluble proteins during the fermentation process, in which the ethanol concentration gradually and continuously increases.

Significant induction of Hsp70 expression was hardly observed during fermentation in SGM. Expression levels of Hsp104 were increased by transferring cells from SD medium to SGM but were lower during the fermentation process than those induced by pretreatment with 6% (vol/vol) ethanol under laboratory conditions. Therefore, the molecular mechanisms of ethanol resistance acquired through fermentation in SGM or pretreatment with 6% (vol/vol) ethanol in SD medium might not be the same. Protein disaggregation by Hsp104 in the bi-chaperone system is triggered by aggregated protein recruitment induced by Hsp70. This series of reactions is ATP dependent (29–32). Conversely, Hsp104 exhibits ATP-independent holdase activity that prevents the aggregation of soluble proteins (40). It is likely more efficient for yeast cells to prevent protein aggregation induced by the gradual ethanol increase through holdase activity of Hsp104 than to disaggregate and regenerate aggregated proteins by enhancing the bi-chaperone system with increased Hsp70 expression. As ATP production efficiency is reduced during fermentation due to the suppression of aerobic respiration, suppressing Hsp70 expression may also be effective in reducing ATP consumption.

The modest increased expression of Hsp104 and near-negligible induction of expression of Hsp70 suggest that other factors may be involved in increasing resistance during the fermentation process. Trehalose inhibits protein denaturation (41, 42) and promotes cell survival under severe ethanol stress (43). Additionally, trehalose levels increase in yeast cells during fermentation (44–46). Therefore, increased trehalose levels likely contributed to the prevention of protein denaturation during fermentation. Autophagy and proteasome-related functions are required for optimal survival of

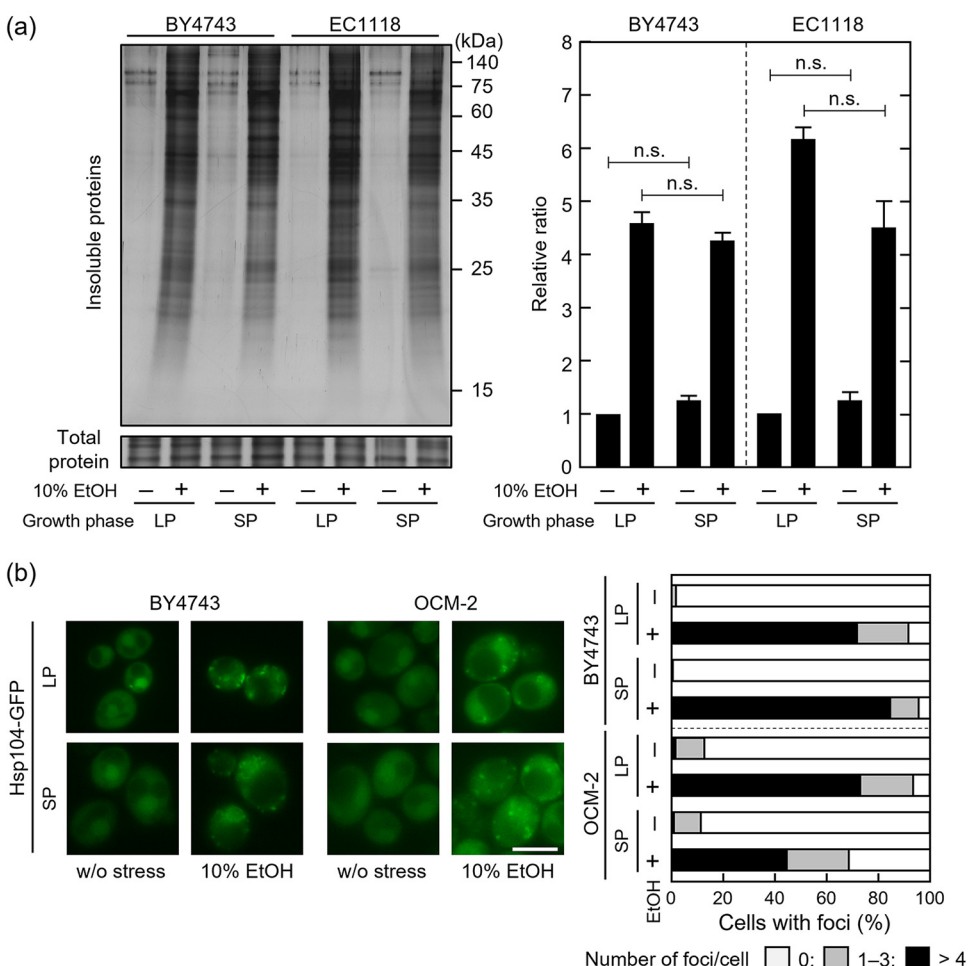

**FIG 7** Stationary-phase (SP) and log-phase (LP) cells accumulated insoluble proteins under severe ethanol stress. Cells cultured in SD medium were treated with or without 10% (vol/vol) ethanol for 180 min. (a) Intracellular insoluble protein levels were assayed. Each value is expressed as the mean ± SE of fold changes in the insoluble protein staining levels relative to LP cells without ethanol stress treatment ($n = 3$). (b) Deposition site formation was examined using Hsp104-GFP. Representative images are shown in the left panels and quantified data are presented in the right panel. A total of 100 cells under each condition were examined, and experiments were conducted in triplicate (300 cells in total were examined). Scale bar = 5 $\mu$m.

wine yeast cells during fermentation (47). Proteolysis has been observed during the fermentation process (48). Autophagy activation has also been reported during the early stages of fermentation when nitrogen is still available (47). These proteolytic systems may remove the insoluble proteins produced during fermentation without delay.

As other factors besides Hsp70/Hsp104, trehalose, and the proteolytic systems may be involved in resistance, further analysis is needed to elucidate the mechanism of acquired resistance during fermentation. Trehalose synthesis and various molecular chaperones depend on the Hsf1 and Msn2/Msn4 stress-responsive transcription factors for transcriptional activation (49–51). Unlike EC1118 and OCM-2, the Japanese sake yeast Kyokai no. 7 (K7) and its relatives do not effectively induce trehalose synthesis and stress responses, as Hsf1 and Msn2/Msn4 are barely activated. This is due to homozygous loss of function of Rim15, an upstream activation factor of Hsf1 and Msn2/Msn4 (52, 53). Similar fermentation studies using K7 and its relatives would help to estimate the importance of Hsf1- and Msn2/Msn4-mediated transcriptional activation, further elucidating the mechanisms of acquired resistance.

Fermentation temperature is a critical parameter that affects ethanol production efficiency and yeast physiology (54, 55). Grape must fermentation at 15°C and 30°C resulted

in differences in fermentation efficiency and intracellular pH maintenance (37). In the present study, fermentation tests were performed at 15°C to mimic the vinification process of white wine, whereas red wine is commonly vinified at temperatures between 20°C and 30°C. The SGM ethanol concentration increased gradually, rather than rapidly, at 15°C. This may have enabled the yeast cells to gradually become resistant to ethanol and prevent the accumulation of insoluble proteins. Sufficient preparation time provided by a gradual increase in ethanol may be necessary to acquire tolerance. Accelerated increase in the grape must ethanol concentration by raising the fermentation temperature may affect ethanol resistance and proteostasis of wine yeast cells.

Additionally, yeast cell physiology changes significantly during winemaking (37, 56–58). Yeast cells reportedly maintain their metabolic activity and survival capacity during the final stage of the winemaking process at 15°C (37). Our results also demonstrated that yeast cells became more robust during fermentation. However, the physiology of yeast cells during winemaking remains unclear, and the various processes involved cannot be clarified through analysis under laboratory conditions. Further studies of the fermentation process will improve our understanding of yeast cell physiology during winemaking. Analysis of the fermentation process at low temperatures, where fermentation proceeds slowly over a long period, will likely provide insights into yeast cell survival and adaptation under severe ethanol stress conditions. It may also provide useful insights to improve winemaking technologies.

## MATERIALS AND METHODS

**Strains and media.** BY4743 (*MAT**a**/α his3Δ1/his3Δ1 leu2Δ0/leu2Δ0 LYS2/lys2Δ0 met15Δ0/MET15 ura3Δ0/ura3Δ0*) was used as the laboratory yeast strain. EC1118 and OCM-2 were used as the wine yeast strains (58–62). EC1118 is a diploid commercial strain and one of the most widely used strains worldwide in winemaking (60). OCM-2 is an isogenic gene-deletion mutant of OC-2 with *ura3Δ/ura3Δ* (61). OC-2 is a diploid and homothallic strain that is highly resistant to sulfurous and high-sugar stress (62). OC-2 is commonly used in winemaking (62). Laboratory experiments consisting of yeast cells were cultured in 50 mL SD medium (2% glucose, 0.67% yeast nitrogen base without amino acids, 20 mg/L L-histidine HCl, 100 mg/L L-leucine, and 20 mg/L uracil) at 28°C with reciprocal shaking (120 rpm) to an optical density at 600 nm ($OD_{600}$) of 0.5 (LP cells). SP cells were prepared through cultivation in 50 mL of SD medium at 28°C to an $OD_{600}$ of 4.0 (BY4743) or 8.0 (EC1118 and OCM-2). Laboratory stress treatment procedures were conducted as previously described (23). The alcoholic fermentation test consisted of wine yeast cells that were prepared through cultivation in 50 mL SD medium at 28°C to and $OD_{600}$ of 1.0. Yeast cells were harvested using centrifugation, transferred to 200 mL of the SGM (33), and fermented at 15°C without shaking. The must ethanol concentration was measured using gas chromatography (AL-2; Riken Keiki Co., Tokyo, Japan).

**Plasmids.** The construction of YIp-*HSP104-GFP* and YIp-*SIS1-GFP* was described in our previous studies (23, 35).

**Protein analysis.** Hsp104 and Hsp70 protein levels were monitored by Western blotting using an anti-Hsp104 (ADI-SPA-1040-D; Enzo Life Sciences, Inc., Farmingdale, NY, USA), anti-Hsp70 (SMC-162c; StressMarq Biosciences, Inc., Victoria, BC, Canada), anti-mouse IgG, horseradish peroxidase (HRP)-linked (7076S; Cell Signaling Technology, Beverly, MA, USA), and anti-rabbit IgG, HRP-linked (7074S; Cell Signaling Technology) antibodies. Ponceau S staining was performed to confirm the equal loading and transfer of all proteins. Insoluble aggregated protein levels were analyzed by silver staining (23, 63). Cells were treated with Zymolyase-20T (2.5 mg/mL) (Nacalai Tesque, Kyoto, Japan) at 25°C and then disrupted by vortexing with glass beads in lysis buffer (50 mM potassium phosphate buffer containing 5% glycerol and 1.0 mM EDTA, pH 7.0). Unbroken cells and debris were removed by centrifugation (200 × *g* for 20 min). The total protein concentration of each sample was normalized. Insoluble aggregated proteins were pelleted by centrifugation (16,000 × *g* for 20 min), washed twice with lysis buffer containing 2% NP-40, and solubilized with urea buffer (50 mM Tris-HCl buffer containing 5% SDS and 6.0 M urea, pH 7.5). The samples were subjected to 10% polyacrylamide gel electrophoresis and visualized by silver staining using Sil-Best Stain One (Nacalai Tesque). Western blotting and silver staining bands were quantified using ImageJ software (http://imagej.nih.gov/ij/) and normalized to Ponceau S staining and total protein, respectively. The significance of the differences was evaluated using unpaired two-tailed Student's *t* test (*, $P < 0.05$; **, $P < 0.01$; n.s., statistically nonsignificant difference).

**Fluorescent microscopy analysis.** An IX83 microscope system (Olympus, Tokyo, Japan) was used for fluorescence microscopy analysis. The percentage of cells containing Hsp104-GFP and Sis1-GFP foci was calculated by examining 100 cells from each experimental group. Experiments were repeated in triplicate (300 cells in total were examined for each condition).

## ACKNOWLEDGMENTS

The present study was supported by the Japan Society for the Promotion of Science under grant numbers 19H02884 and 20H02900 and by the Ohsumi Frontier Science Foundation to S.I.

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
