## [Reviewer comments · Microbiology Spectrum]

Microbiology Spectrum

Wine yeast cells acquire resistance to severe ethanol stress and suppress insoluble protein accumulation during alcoholic fermentation

Masashi Yoshida, Noboru Furutani, Futsuki Imai, Takeo Miki, and Shingo IZAWA

Corresponding Author(s): Shingo IZAWA, Kyoto Institute of Technology (K.I.T.)

Review Timeline:

Submission Date:	March 16, 2022
Editorial Decision:	May 15, 2022
Revision Received:	July 27, 2022
Accepted:	August 14, 2022

Editor: Kate Howell

Reviewer(s): Disclosure of reviewer identity is with reference to reviewer comments included in decision letter(s). The following individuals involved in review of your submission have agreed to reveal their identity: Agustin Aranda (Reviewer #1)

Transaction Report:

DOI: <https://doi.org/10.1128/spectrum.00901-22>

May 15, 2022

Dr. Shingo IZAWA
Kyoto Institute of Technology (K.I.T.)
Laboratory of Microbial Engineering
Gosho-kaido-cho
Matsugasaki, Sakyo-ku
Kyoto, Kyoto 6068585
Japan

Re: Spectrum00901-22 (**Wine yeast cells acquire resistance to ethanol-induced protein denaturation during alcoholic fermentation**)

Dear Dr. Shingo IZAWA:

We have had two expert reviewers look at your paper and while some merit was found there needs to be work done on the paper. Please pay close attention to the comments, particularly regarding precision of terms, and controls in experiments.

Link Not Available

Sincerely,

Kate Howell

Journals Department
Reviewer comments:

Reviewer #1 (Comments for the Author):

The manuscript "Wine yeast cells acquire resistance to ethanol-induced protein denaturation during alcoholic fermentation" by Yoshida et al. is an interesting analysis of ethanol-induced protein aggregation in the context of industrial strains and winemaking conditions. Little is known of those mechanisms of quality control operating in conditions of biotechnological interest. The results are also interesting from the basic point of view, as it is demonstrated that ethanol does not trigger a strong protein

accumulation when its accumulation is slow, so cells have mechanism to prevent those kind of events.

It is clear that there is no a strong protein aggregation during winemaking compared to a high ethanol shock. However, it does not mean that there is no stress in grape juice fermentation. In the beginning of fermentation there is a hyperosmotic shock and low pH. That would explain the relatively high levels of Hsp70 at day 1 of fermentation (Figure 4) compare to non-stress situation (Figure 2). Day 1 of fermentation is not a good reference point, so a Western containing both samples from fermentation and samples on exponential log-phase of growth in SD medium has to be included to show the real degree of stress response. The same applies for silver staining of protein aggregates t in Figure 3b that is key to show lack of protein aggregates in winemaking conditions. Levels of aggregation are lower than the control with an ethanol shock, but protein samples from exponential cells (Figure 1) seem to have a lower degree of band intensities. Both samples have to be included together in the same gel to get a meaningful result.

The experiments of silver staining of protein aggregation have to be fully explained in the Materials and Methods section, as are key experiments not widely used. Explain the method of protein extraction in these experiments. For fluorescence experiments, Hsp104 has been used as marked of aggregation during winemaking. As it is a protein involved in the removal of these aggregates, wouldn't it be better to use proteins involved in aggregation such Hsp42, Btn2, and Sis1 as reporters? Are the results the same in other conditions?

Minor points

Line 100. Explain that OCM-2 is just a *ura3* auxotrophic EC1118 strain for cloning purposes, not a different commercial strain
Line 206. It is far too speculative to say that Hsp70 and Hsp104 are the main factors influencing acquired resistance during the fermentation process in any condition. Stress response is very wide and complex.

Reviewer #3 (Comments for the Author):

In this study, Yoshida et al. investigate the accumulation of proteins in an insoluble fraction by western blot as an indicator of protein denaturation and aggregation caused by ethanol stress. In some instances, findings are further supported by imaging cellular deposits of the chaperone Hsp104 that has been fluorescently tagged. The data presented in this paper are of good quality and generally well controlled; however, issues to address are outlined below.

1. The data presented supports the idea that (1) wine yeast respond to ethanol stress like laboratory strains and (2) the progression of wine yeast through fermentation in SGM results in stress tolerance, which is similar to pre-treatment of lab strains with 6% EtOH or heat. In the text, including the importance paragraph, this is not clearly presented. For example, on line 46 it fails to note that it is acute stress that is being referred to and the response is the same in wine yeast, while lines 47-50 seem to indicate the response is different. Lines 51-53 is also overstated since this is known yeast physiology that is shown to be the same in winemaking. I would like to see text throughout the manuscript changes to reflect the two points above.
2. The authors switch back and forth between talking about insoluble and denatured protein. There is no data to directly assess the folding state of isolated proteins. As such, language in the paper should be changed to indicate that they are measuring the insoluble protein fraction (really a protein fraction that pellets in their assay). Given the ability of ethanol to denature proteins, it can be suggested that this is an indication of protein denaturation; however, this cannot be known for sure given the data.
3. I recommend that authors reconsider how they discuss data in figure 4. It appears that on day 1 Hsp70 is induced, as the levels look different than what is seen in SD without stress. This may indicate early stress in SGM that primes the cells and lasts throughout fermentation, like pre-conditioning does with 6% EtOH or heat. Earlier times in SGM and a SD sample on the same western blot would help with this interpretation. Similarly, it looks like Hsp104 is higher than in SD without stress after day 1. As such, I suspect the cells are mildly stressed early in fermentation to have elevated PQC activity that can handle the amount of denatured protein being generated over time in the fermentation. In other words, acute stress causes much higher PQC expression levels due to the immediate generation of a lot of denatured material at once.
4. I suggest the authors change their wording around the idea of cells and proteins becoming "resistant to denaturation". See importance statement and line 174 as examples. First it is not cells that are denaturing, it is their cellular components. Second, with respect to protein denaturation, I do not think proteins are becoming resistant to denaturation. Rather the protein machinery needed to refold proteins is being activated to counteract ethanol induced denaturation and mis-folding. It would be OK to say that cells are becoming resistant to the cellular impact of ethanol via induction of the PQC system.
5. The authors should not use the term brewing in their manuscript, as this is commonly used to refer to the making of beer. I would replace this with fermentation or wine making.

Staff Comments:

Preparing Revision Guidelines

Please return the manuscript within 60 days; if you cannot complete the modification within this time period, please contact me. If you do not wish to modify the manuscript and prefer to submit it to another journal, please notify me of your decision immediately so that the manuscript may be formally withdrawn from consideration by Microbiology Spectrum.

Response letter

Thank you to all the reviewers for your time and constructive comments on our manuscript (Spectrum00901-22). We have revised the manuscript accordingly. The revised parts in the manuscript were written in red. We sincerely hope that the revised version is acceptable.

To Reviewer #1

We appreciate your comments regarding our study. We have revised our manuscript accordingly.

1. *“Day 1 of fermentation is not a good reference point, so a Western containing both samples from fermentation and samples on exponential log-phase of growth in SD medium has to be included to show the real degree of stress response. Levels of aggregation are lower than the control with an ethanol shock, but protein samples from exponential cells (Figure 1) seem to have a lower degree of band intensities. Both samples have to be included together in the same gel to get a meaningful result.”*

Response: In response to your comment, we ran samples from the fermentation process and the logarithmic growth phase in SD medium on the same gels (E0, E10, and E6/10 in Figs. 3 and 4), which confirmed that, throughout the fermentation process, wine yeast cells maintained insoluble proteins and Hsp70 at levels nearly the same or lower than those of cells cultured in SD medium without ethanol.

2. *“The experiments of silver staining of protein aggregation have to be fully explained in the Materials and Methods section, as are key experiments not widely used. Explain the method of protein extraction in these experiments.”*

Response: We have added a description of the protein extraction method (Lines 300-314).

3. *“As Hsp104 is a protein involved in the removal of these aggregates, wouldn't it be better to use proteins involved in aggregation such Hsp42, Btn2, and Sis1 as reporters? Are the results the same in other conditions?”*

Response: We observed Sis1-GFP localization in OCM-2. Sis1-GFP also formed foci under acute ethanol stress in SD medium. However, foci formation was suppressed during the fermentation process (Figs. 1 and 3).

Minor points

“Explain that OCM-2 is just a ura3 auxotrophic EC1118 strain for cloning purposes, not a different commercial strain.”

Response: OCM-2 was derived from another commercial strain, OC-2. OCM-2 is not an *ura3* auxotrophic EC1118 strain. We have added the background and references for OC-2 and EC1118 to the revised manuscript (Line 273-277).

“Line 206. It is far too speculative to say that Hsp70 and Hsp104 are the main factors influencing acquired resistance during the fermentation process in any condition. Stress response is very wide and complex.”

Response: Following your suggestion, we removed the relevant parts of the manuscript.

To Reviewer #3

Thank you for your constructive comments that have markedly improved the quality of our manuscript. We have revised our manuscript according to your suggestions as described below.

1 *“The data presented supports the idea that (1) wine yeast respond to ethanol stress like laboratory strains and (2) the progression of wine yeast through fermentation in*

SGM results in stress tolerance, which is similar to pre-treatment of lab strains with 6% EtOH or heat. In the text, including the importance paragraph, this is not clearly presented. For example, on line 46 it fails to note that it is acute stress that is being referred to and the response is the same in wine yeast, while lines 47-50 seem to indicate the response is different. Lines 51-53 is also overstated since this is known yeast physiology that is shown to be the same in winemaking. I would like to see text throughout the manuscript changes to reflect the two points above.”

Response: Based on your comments, we have revised the Importance and Discussion sections of the manuscript.

2. “The authors switch back and forth between talking about insoluble and denatured protein. As such, language in the paper should be changed to indicate that they are measuring the insoluble protein fraction.”

Response: Following your suggestion, we have revised the manuscript using only the term “insoluble proteins.”

3. “I recommend that authors reconsider how they discuss data in Figure 4. It appears that on day 1 Hsp70 is induced, as the levels look different than what is seen in SD without stress.”

Response: We compared the expression levels of Hsp70 and Hsp104 multiple times by running samples from the fermentation process and the logarithmic growth phase in SD medium on the same gels (Figs. 3 and 4). We observed that Hsp70 levels increased slightly on day 1 in EC1118 cells, and then decreased to almost basal levels. However, the induction of Hsp70 expression was near-negligible throughout the fermentation process. We also confirmed that Hsp104 expression levels in cells during the fermentation process were higher than those in cells in SD medium without stress, but lower than those in cells treated with ethanol.

4. *“I suggest the authors change their wording around the idea of cells and proteins becoming "resistant to denaturation". It would be OK to say that cells are becoming resistant to the cellular impact of ethanol via induction of the PQC system.”*

Response: Thank you for your comment. Following your comment, we have revised the title and sentences related to the manuscript.

5. *“The authors should not use the term brewing in their manuscript, as this is commonly used to refer to the making of beer. I would replace this with fermentation or wine making.”*

Response: We replaced the word “brewing” with “wine making” or “fermentation.”

August 14, 2022

Dr. Shingo IZAWA
Kyoto Institute of Technology (K.I.T.)
Laboratory of Microbial Engineering
Gosho-kaido-cho
Matsugasaki, Sakyo-ku
Kyoto, Kyoto 6068585
Japan

Re: Spectrum00901-22R1 (**Wine yeast cells acquire resistance to severe ethanol stress and suppress insoluble protein accumulation during alcoholic fermentation**)

Dear Dr. Shingo IZAWA:

Your manuscript has been accepted, and I am forwarding it to the ASM Journals Department for publication. You will be notified when your proofs are ready to be viewed.

Sincerely,

Kate Howell
Editor, Microbiology Spectrum
